# Duodenal Gangliocytic Paragangliomas—Case Series and Literature Review

**DOI:** 10.3390/life13030597

**Published:** 2023-02-21

**Authors:** Madalina Stan-Ilie, Vasile Şandru, Oana-Mihaela Plotogea, Ecaterina Rînja, Christopher Pavel, Gabriel Constantinescu, Lucian Negreanu, Dan Nicolae Paduraru, Alexandra Bolocan, Octavian Andronic, Dragos Davitoiu, Gheorghe G. Bălan, Alexandru Constantinescu

**Affiliations:** 1Department of Gastroenterology, Clinical Emergency Hospital of Bucharest, 105402 Bucharest, Romania; 2Department 5, “Carol Davila” University of Medicine and Pharmacy, 050474 Bucharest, Romania; 3Department of Gastroenterology, University Emergency Hospital Bucharest, 050098 Bucharest, Romania; 4Department of Surgery, University Emergency Hospital Bucharest, 050098 Bucharest, Romania; 5Department of Gastroenterology, “Grigore T. Popa” University of Medicine and Pharmacy, 700115 Iași, Romania

**Keywords:** duodenal gangliocytic paraganglioma, jaundice, gastrointestinal bleeding, endoscopy, surgery, immunohistochemistry

## Abstract

Duodenal gangliocytic paragangliomas are rare neuroendocrine tumors primarily localized in the periampullary area. Though mostly asymptomatic, they can present with various symptoms, most often jaundice, anemia and abdominal pain. The present paper is a case series report, describing our personal experience with patients presenting to the Emergency Unit with different symptoms due to duodenal gangliocytic paraganglioma. Endoscopic resection is safe and indicated in most of the cases, being also associated with lower medical costs. EUS plays a central role in the pre-resection management and in surveillance, and immunostaining is decisive to ascertain the tumor histologic origin. In addition to reporting our experience, we researched the literature regarding these rare tumors and performed a comprehensive review.

## 1. Introduction

Duodenal gangliocytic paragangliomas (GP) are rare neuroendocrine tumors, with less than 300 cases being reported worldwide, most of them as single case reports. Duodenal GPs arise mainly in the second part of the duodenum and in the periampullary region. Unlike other neuroendocrine neoplasia, GPs have a benign clinical course, being rather asymptomatic and sometimes incidentally found during imaging studies or autopsy [1,2]. When they rarely present clinical manifestations, these include upper gastrointestinal bleeding due to mucosal ulceration, abdominal pain, anemia and seldomly biliary obstruction [3]. Exceptionally, GPs may recur after resection or may metastasize in nearby lymph nodes. However, most cases are treated only by local excision, without subsequent lymph node investigation. Therefore, the incidence of metastasis may be underestimated [4].

Due to their low incidence and vague presentation, their diagnosis could prove difficult without a skilled eye. Awareness about their clinical and endoscopical presentation, as well as their histological features should be raised. In this paper, the authors of the study report their experience with three patients that presented with different clinical manifestations of duodenal GPs, which were eventually managed endoscopically or surgically during the period of one year. The main aim is to highlight inapparent clinical, endoscopic diagnostic features of the pathology as well as its therapeutic management. Moreover, we searched multiple databases (PubMed, EMBASE, Cochrane Central) for studies reporting the presence of GP and realized a literature review on this topic. Following the international guidelines, our study could add to a relatively scantily represented segment of medical literature, mainly in the field of patient management and diagnosis [5].

## 2. Case Report 1

We present the case of a 29-year-old male, without any significant medical history except for first grade obesity, who was sent to our hospital for recently debuted sclero-tegumentar jaundice, pruritus and epigastric pain, with a preliminary tomographic diagnosis of both vesicular lithiasis and choledocholithiasis established in another medical unit. Biochemically, no anemia or leukocytosis were noted, and outside of slight cholestasis and hepatocytolysis, no other changes were identified. Given the young age, we initially directed him toward endoscopic ultrasonography (EUS) for confirmation of the diagnosis. Later on, we sent the patient to an endoscopic retrograde cholangiopancreatography (ERCP) procedure for papillosphincterotomy and stone extraction. After carefully examining the ampullary region, we noticed a slightly elevated, inapparent, tumor-shaped protrusion adjacent to the major papilla, with a normal mucosal lining. A differential diagnosis was made between a benign tumor, an eerie biliary stone impaction or a bulge of mucosa resembling a tumor. Therefore, we set for cannulation and biliary tract contrast injection, which showed, along with two lacunar round images suggestive of calculi, an enlarged common bile duct (CBD) of up to 10 mm and a distal biliary stenosis in the last 15 mm. Stone extraction was performed using a basket, and a 10 Fr/7 cm stent was placed (Figure 1).

We further carried out a second EUS examination with a special focus on the region, and a well-delineated 14/7 mm hypoechoic, homogeneous, submucosal mass in the periampullary region of the second duodenum was identified. The tumor had Doppler signal and presented a mixed elastographic pattern. The lesion was punctured under EUS guidance by means of a 22G Franseen-tip biopsy needle (Acquire^TM^, Boston Scientific Corporation, Marlborough, MA, USA), three passes being conducted through the tumor (Figure 2).

The samples were first fixed in a 10% formalin solution and paraffin-embedded (formalin-fixed paraffin-embedded—FFPE); afterwards, micrometer-thick sections were sliced at a microtome. Morphological studies using Giemsa and Hematoxylin–Eosin (HE) colorations were carried out, and then immunostaining was performed. Microscopically, the tumor cells were ovoid, monomorphic, presenting mild nuclear atypia, low mitotic activity and no necrosis. As for the immunohistochemistry staining, the fragments were positive for synaptophysin, chromogranin A and CD56 and focally positive for pan-cytokeratin AE1/AE3. They stained negative for S100, CD34, MSA. The Ki67 score was 1%. Finally, the pathologist classified the lesion as a neuroendocrine well-differentiated tumor (NET grade 1).

Afterwards, the patient underwent further imaging studies, computed tomography (CT) and magnetic resonance imaging (MRI), for preoperative stadialization, which rendered the tumor resectable (Figure 3). Therefore, we proceeded with surgical management in the form of radicality—duodenotomy, tumoral resection, stent extraction, duodenoraphy, peritoneal drainage and then placement of a nasojejunal tube for enteral nutrition, with good postoperative prognosis.

Morphological studies of the whole resected tumor showed a nodular, well-defined structure, covered with duodenal lining with clear resection margins. Microscopically, it featured three-phasic tumoral proliferation consisting of Schwannian-type spindle-shaped cells, ganglia-like cells and neuroendocrine-type cells with nests, acini and trabeculae; rare glandular structures were noted. These findings were consistent with the diagnosis of gangliocytic paraganglioma. The recovery after the surgical intervention was excellent and, after one year of regular follow-up, the patient has no tumoral recurrence and no symptoms (Figure 4 and Figure 5).

## 3. Case Report 2

The second patient was a 43-year-old male, with a medical history of primary arterial hypertension and obesity class 1, referred to our Emergency Unit for black, tarry stools, known as melena, suggestive of upper gastrointestinal bleeding, epigastric pain, hypotension and asthenia. These symptoms started 7 days before presentation. Biologically, a hemoglobin value as low as 5 g per deciliter was detected. The patient was hemodynamically stabilized with blood transfusions and plasma expanders and also received high-dose proton pump inhibitor therapy. An upper gastrointestinal endoscopy was performed, and in the second part of the duodenum, in the nearby vicinity of the major and minor duodenal papilla, an extramucosal, ulcerated, polipoid lesion of about 20/15 mm was noted, with a hematic clot covering the ulcer base and a diffuse slow hemorrhage noted at the clot’s extremities. At first, hemostasis was conducted by perilesionally injecting 20 mL of adrenaline 1:10,000. After this, we performed a CT exam that excluded the possible local or distant spread of the tumor. The next day, the upper endoscopy by means of a side-viewing duodenoscope was repeated with the intention to remove the tumor (Figure 6).

Therefore, under Propofol sedation, the patient underwent endoscopic resection with the use of a 25 mm polypectomy snare. Therefore, we injected diluted adrenaline (1:10,000) into the nearby duodenal wall, and the tumor was resected into two pieces with dimensions of 20/15/5 mm and 15/10/5 mm, respectively (Figure 7).

Hemostasis by means of an adrenalin 1:10,000 injection and electrocoagulation with a 10 Fr bipolar probe was used to minimize the risk of bleeding (Figure 8).

Microscopically, we observed tumoral proliferation in three components: spindle-shaped cells, which vastly outnumbered the others, focal ganglia-type cells and areas with medium-sized epithelioid cells with a neuroendocrine phenotype, disposed in a tubulo-papillar fashion. There was no obvious mitotic activity, and no necrosis was detected. The overlying duodenal mucosa, as well as the peripheral tissue contained no tumor elements (Figure 9).

The immunochemistry analysis showed S100 and synaptophisine diffusely staining, as well as chromogranin staining in the epithelioid tumor component, which was also partially stained for EMA, AE1/AE3 and PGR. The Ki67 score was 2–4% (Figure 10).

The histopathological results and immunochemical assays were consistent with the diagnosis of gangliocytic paraganglioma.

There were no periprocedural complications, and the patient was discharged after 5 days of surveillance. Six months later, the patients underwent endoscopic and endosonographic evaluation, which prompted no further therapeutic steps, the resection spot being barely visible, with no recurrence signs by EUS.

## 4. Case Report 3

The third patient was a 41-year-old male, without any relevant medical history outside of obesity class 1 and essential hypertension, presenting to endoscopic evaluation for symptomatic reflux disease and abdominal pain. The blood tests only showed a slightly elevated creatininemia and minimal anemia, with no other pathological changes, including tumoral markers detection. The Helicobacter pylori stool antigen test rendered a positive result. Abdominal ultrasound scanning yielded normal results. At first, the patient underwent endoscopy by means of a standard frontal-viewing endoscope and, a 4 mm sessile polyp Paris Is was identified in the bulbar area, close to the pyloric orifice, hidden by the posterior sphincterian wall (Figure 11). Biopsies were taken, but the histopathological studies yielded no relevant result, outside of the presence of a chronic inflammatory infiltrate, without any dysplastic or neoplastic cells. Also MRI did not identify the duodenal nodule but issued images of enlarged hepatic hilar and retropancreatic lymphadenopathies, with somewhat significance in the context.

The next logical step was preparing for polypectomy, and 3 days later we examined the lesion again with a frontal-viewing endoscope which visualized the aforementioned, inapparent lesion, but we were unable to attain an optimal resection position of the endoscope. Therefore, we withdrew the standard endoscope and entered the duodenum with a lateral-viewing duodenoscope, which offered a much better strategic position (Figure 12A). Electroexcision was accomplished with the use of a polypectomy snare after injecting the duodenal wall with methylene blue (Figure 12B). The entire polyp was retrieved with a basket and sent to our pathologist who rendered a gangliocytic paraganglioma diagnosis of submucosal origin (Figure 12C,D).

A morphological analysis showed cells with hyperchromic nuclei, elongated, spindle-like or round-oval in shape, with a low cytoplasmic volume, alternating with cells with vesicular nuclei and a high-volume, clear, granular cytoplasm. Lymphoid tissue was present in an intertwined fashion with the tumoral tissue (Figure 13). Necrotic tissue was absent, and a low mitotic activity was observed. Immunostaining revealed positivity for synaptophisin, chromogranin, AE1/AE3. CDx2 staining was negative. The Ki67 score was positive in a mere 1–2% of the cells.

The patient was monitored shortly after the procedures and on the long run, endoscopy being performed after 3, 6 and 12 months, revealing complete local healing in a normal clinical context, outside of a mild persistent abdominal pain managed by proton pump inhibitors, non-steroidal anti-inflammatory drugs and pain medication. A rapid urease test was performed, with negative results due to eradication therapy that the patient was offered meanwhile. Furthermore, endoscopic ultrasound was conducted shortly after the resection and 12 months after, in order to confirm complete resection.

Outside of the persistent abdominal pain, which seemed to be functional, another persistent feature, important in the context, was the continuation of arterial hypertension at a certain degree; so our patient was sent for Holter monitoring of blood pressure for 24 h, which showed a range of systolic blood pressure of 113–144 mmHg and a range of diastolic blood pressure of 65–101 mmHg. The patient was prescribed a calcium channel blocker and sent for continuation of investigations. Serum chromogranin A, gastrin or serotonin (5-hydroxytryptamine) and urinary catecholamines, vanillylmandelic acid or 5-hydroxyindoleacetic acid were tested and yielded normal results. Later on, even positron emission tomography (PET) with radiolabeled somatostatin analogs and a neuroendocrine tumor protocol were carried out, with no gastro–duodenal, bilio–hepato–pancreatic, retroperitoneal or adenopathic anomalies detected (Figure 14).

## 5. Discussions and Literature Review

Pheochromocytomas (PCCs) and gangliocytic paragangliomas (GPs) are rare abnormal growths of chromaffin cells in the adrenal glands and elsewhere in the body, respectively. They can occur in almost any location, except for the bones and brain, all tumors having metastatic potential, with 20% of all tumors presenting in at least two primary locations (multicentric gangliocitic paraganglioma) [6]. Their hereditary predisposition is well documented, a germline mutation being present in almost 40% of tumors and being associated with various syndromes and familial conditions. Genetic testing should be offered to all patients, and patient genetic status represents a key element for diagnosis, prognosis and follow-up [7].

The first-ever description of a duodenal GP by Dahl in 1957 as a duodenal ganglioneuroma was soon followed by the account of Taylor and Helwig in 1962 of a benign nonchromaffin paraganglioma [8,9,10]. Gangliocytic paraganglioma was the term primarily coined by Zacharias and Kepes to describe these duodenal tumors, for their paraganglioma and ganglioneuroma appearance [9]. The most recent World Health Organization classification of tumors of the digestive system defines GPs as “neuroendocrine neoplasms” [11]. Literature data describe a male preponderance, also suggested by our study’s data [12]. Usually, GP is diagnosed as an incidental finding on sectional imaging, by endoscopic studies, or by autopsy, being rather asymptomatic. GPs can rarely present with abdominal pain or jaundice due to mechanical distal biliary obstruction, with only five cases described so far in the literature, and slightly more frequently with upper gastrointestinal bleeding or anemia due to mucosal ulceration [4,12,13,14,15,16,17,18,19,20,21,22,23]. Out of 51 GPs cases included in the study of Burke and Helwig, only one presented with biliary obstruction [12]. However, there are few literature studies evaluating GPs presenting with these particular symptoms. Gangliocytic paraganglioma, which expresses a pathological hallmark consisting in the presence of three main cell lineages—epithelioid, ganglionar and spindle-like cells—mainly occurs in the second part of the duodenum, more frequently in the periampullary area, with much rarer jejunal or pyloric presentation and extremely rare presentations in the esophagus, pancreas or appendix [4,24,25]. Its histogenesis has stirred many theories through the time, but it remains ill-defined. Theories purport that the origin of GP’s epithelioid cells may either be the ventral pancreatic primordium of the endoderm, with a neuroectodermal lineage for the spindle and ganglion cells, or the celiac ganglia or neural crest pluripotent stem cells in the Lieberkühn’s glands, which both arise from the ectoderm [11,26,27]. Usually non-cancerous, GPs can rarely become malignant and metastasize to regional lymph nodes or to other parts of the body, with a single report of a tumor-associated death [28]. Interestingly, in a study of Ogata et al. from 2011, five out of nine patients with metastatic GP to regional lymph nodes presented all three cell lines in histopathologic studies [29]. Radical surgery, though rarely needed, is indicated by the presence of aggressive behavior indicators: infiltrative margins, high mitotic activity and nuclear polymorphism [25]. When factors such as tumoral extension and location seem favorable, endoscopic management should be taken into account [29,30]. Late recurrence through lymph node metastasis has been described, especially for tumors extended beyond the submucosal threshold [31,32].

There is limited research on the use of anti-angiogenic drugs in the treatment of GPs, but some studies have shown promising results of this treatment in combination with conventional therapies. Combining anti-angiogenic therapy and immunotherapy is an emerging strategy in the treatment of gangliocytic paragangliomas. However, more research is needed to determine the optimal combination and dosing of these therapies for gangliocytic paraganglioma [33].

Our tripartite study scoped patients with three particular clinical modes of presentation of GPs: jaundice and abdominal pain, anemia and upper gastro-intestinal bleeding, and symptoms resembling a possible carcinoid syndrome, all tumors being located into the first and second part of the duodenum, i.e., right after the posterior pyloric sphincterian wall, in the nearby vicinity of the minor duodenal papilla, and adjacent to the major papilla.

Interestingly, the demographic and history features were somewhat similar—all patients were young men (29, 41, 43 years old) and were from at least overweight to obese grade I (BMI of 29.9, 30.3 and 30.6). Two of the patients also presented primary arterial hypertension, which in the circumstances of a possible hormone-secreting tumor, may raise questions about the etiology of the raised blood pressure, but the context of our three case reports was not linked to the tumoral manifestations spectrum.

Regarding the diagnostic algorithm, endoscopic ultrasound plays a central role in the early management of every polypoid duodenal tumor, especially for those covered with normal mucosal lining, which can trick the endoscopist into identifying them as normal mucosal bulges, duodenal hyperplastic polyps, diffuse duodenal lymphoid hyperplasia or other common or rare lesions. EUS has the ability of visualizing the origin layer or allowing the tissue acquisition through EUS-FNA or EUS-FNB, very important to obtain a fast diagnosis or in post-endoscopic resection follow-up. Both surgical and endoscopic resection play separate important roles in the treatment of GPs, as our study showed, the latter being advantageous for tumors with superficial location and when local or distant extension is not detected in sectional studies, also lowering the total costs of these patients’ management. Extensive endoscopic and imaging evaluation, as well as biological special tests, are to be done before endoscopic or surgical resection is planned. After this, extensive morphological and staining analyses are needed to ascertain the tumoral origin and differentiate the tumor from neuroendocrine tumors or other benign or malignant nodular lesions. On the long run though, endoscopic, EUS, possibly bioptic, and imaging follow-up are necessary for excluding tumoral relapse, which will raise the costs, despite the tumor’s mainly benign course.

Our three cases highlight another warning sign for complete and exhaustive endoscopic duodenal exploration with both frontal-viewing and lateral-viewing endoscopes when such pathology is suspected.

GPs can manifest with diverse clinical and endoscopic presentation, but they also have some common features, such as demographic or history characteristics—male gender and obesity—and post-resection prognosis, which is much more favorable compared to that of other duodenal tumors, with predominantly excellent survivability and curability. Even though recurrence and metastasis cannot be completely ruled out, a careful preoperative assessment should be conducted [15,16].

Limitations of the study include the lack of genetic testing as well as of the application of a more personalized approach. The lack of long-term monitoring can represent a further limitation.

## 6. Conclusions

Gangliocytic paragangliomas have a polymorphic clinical presentation. These tumors should be suspected in the presence of a polypoid duodenal lesion associated with obstructive jaundice which seems benign by sectional studies, upper digestive hemorrhage and anemia, or symptoms of carcinoid syndrome. EUS plays an important role in the pre-resection management and in surveillance. At the same time, immunostaining is decisive to ascertain a tumor’s histologic origin. Endoscopic resection is safe and indicated in most cases, being associated with few complications, a quick recovery and low medical costs. Surgical resection is recommended when endoscopic resection is unsafe, the tumor is invading the profound layers or metastasis in the lymph nodes are present.

## Figures and Tables

**Figure 1 life-13-00597-f001:**
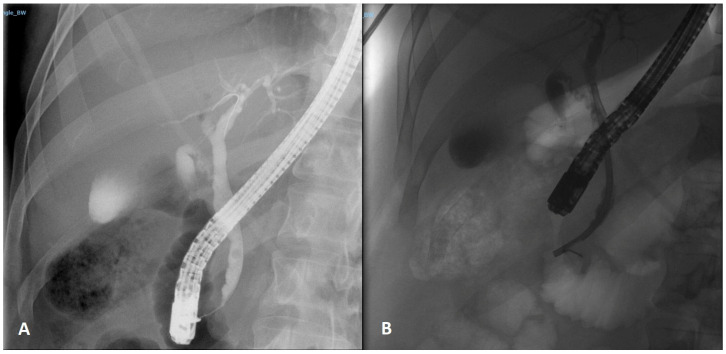
ERCP image of the calculus identified by EUS and of the distal common bile duct stenosis (**A**) through which a plastic stent was inserted (**B**).

**Figure 2 life-13-00597-f002:**
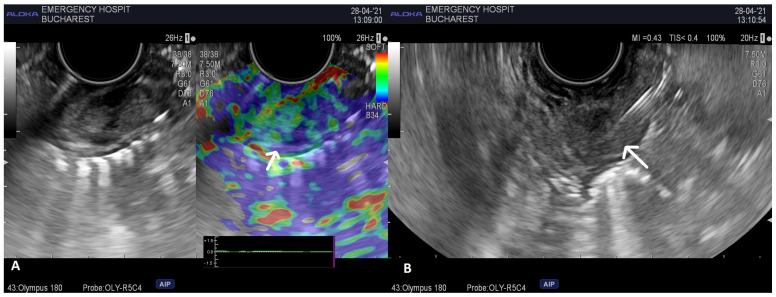
(**A**) EUS image revealing a submucosal lesion in the second part of the duodenum, mixed at elastography (arrow-mixed pattern of the lesion). (**B**) EUS guided-biopsy through EUS-FNB (arrow-needle inside the lesion).

**Figure 3 life-13-00597-f003:**
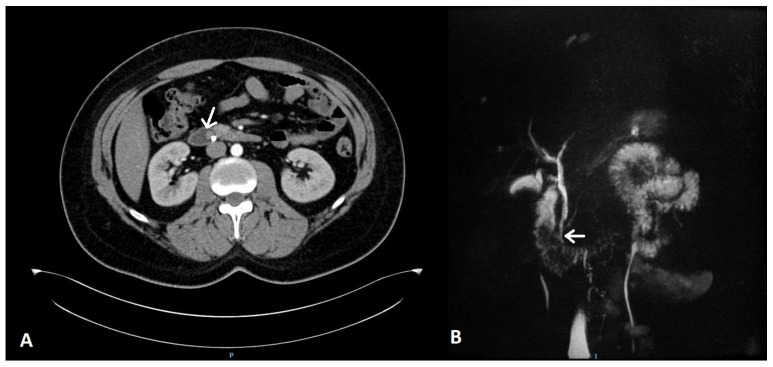
CT (**A**) and MRI (**B**) aspects of the patient in the presurgical step. Arrow—duodenal location of the tumor.

**Figure 4 life-13-00597-f004:**
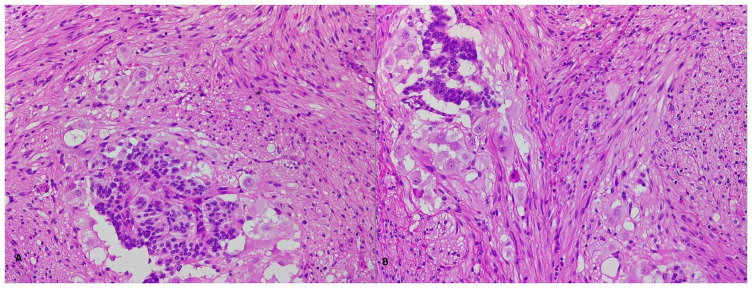
HE-stained examination of tumoral proliferation typical of GP, showing fusiform, epithelial and ganglionar cells (**A**,**B**, 400×).

**Figure 5 life-13-00597-f005:**
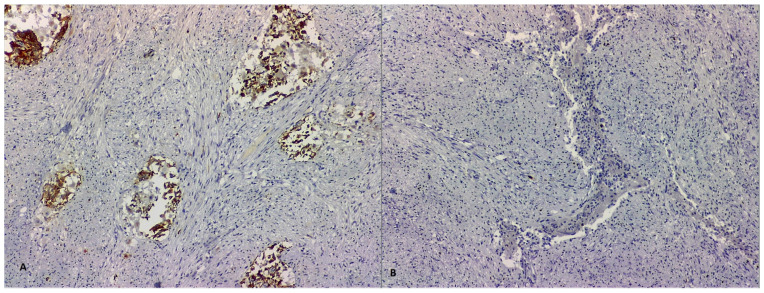
Microscopic immunohistochemical analysis showing chromogranin-positive staining of epithelial cells, but negative staining of fusiform and ganglionar cells (**A**, 100×), along with very rare, Ki67-positive tumoral cells (**B**, 100×).

**Figure 6 life-13-00597-f006:**
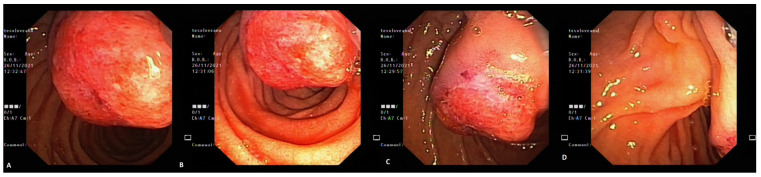
Tumoral inspection with the side-viewing duodenoscope (**A**–**C**) and bile flowing through the papilla right next to the tumor (**D**).

**Figure 7 life-13-00597-f007:**
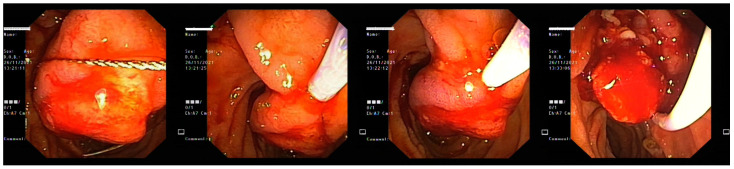
Endoscopic view during tumoral resection.

**Figure 8 life-13-00597-f008:**
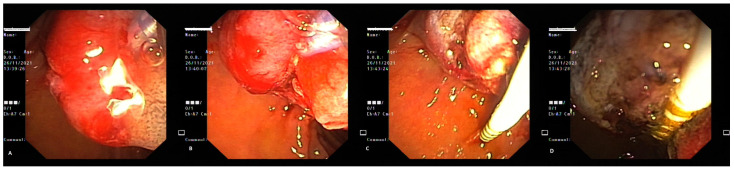
Endoscopic mixed hemostasis for the endoscopic resection site: injection of adrenaline 1:10,000 (**A**,**B**) and bipolar hemostasis (**C**,**D**).

**Figure 9 life-13-00597-f009:**
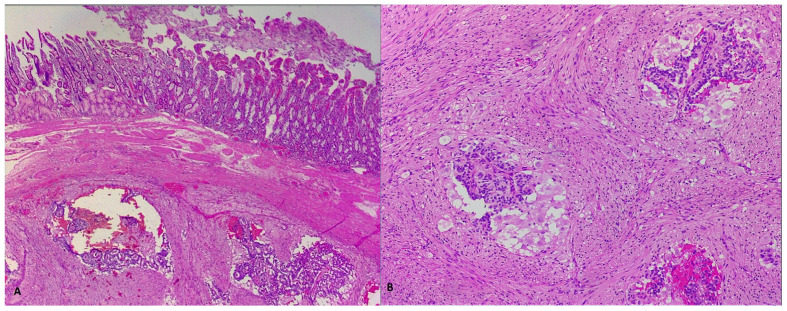
Morphological and HE-stained samples to examine tumoral proliferation typical of GP covered by normal duodenal layers (**A**, 50×) and showing fusiform, epithelial and ganglionar cells (**B**, 200×).

**Figure 10 life-13-00597-f010:**
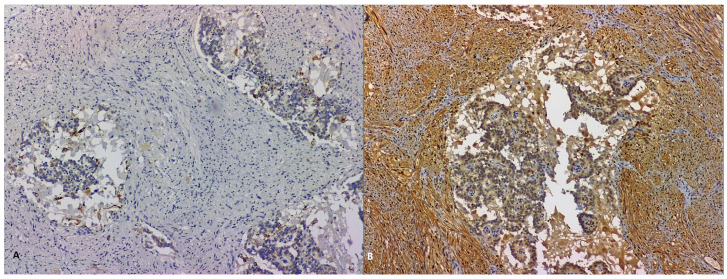
Immunohistochemical analysis of the identified GP tumor showing pan-cytokeratin-focally positive epithelial cells (**A**, 100×) and S100-positive fusiform cells (**B**, 100×).

**Figure 11 life-13-00597-f011:**
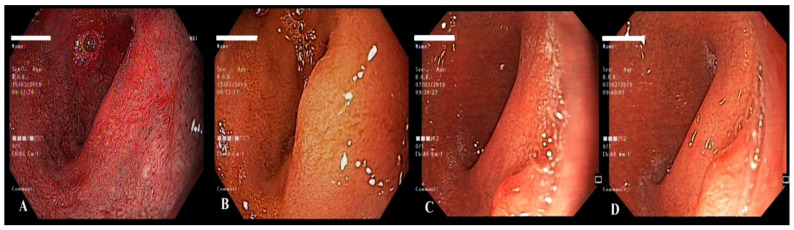
Polypoid lesion in the bulbar portion of the duodenum, right behind the pyloric sphincterial wall, inapparent at first glance (**A**,**B**), identified after a complete endoscopic exam (**C**,**D**).

**Figure 12 life-13-00597-f012:**
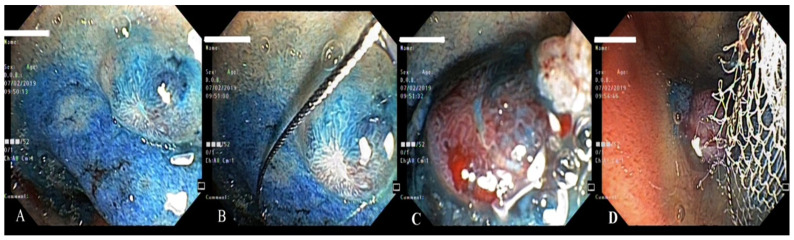
Examination by a lateral-viewing duodenoscope (**A**), electroexcision (**B**,**C**) and retrieval of the polyp (**D**).

**Figure 13 life-13-00597-f013:**
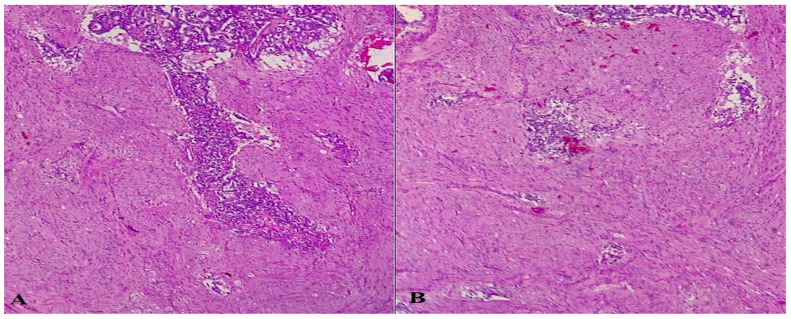
Morphological and standard coloration aspects of the tumor showing large ganglionar tumor cell proliferation (**A**, 100×) and fusiform cells (**B**, 100×).

**Figure 14 life-13-00597-f014:**
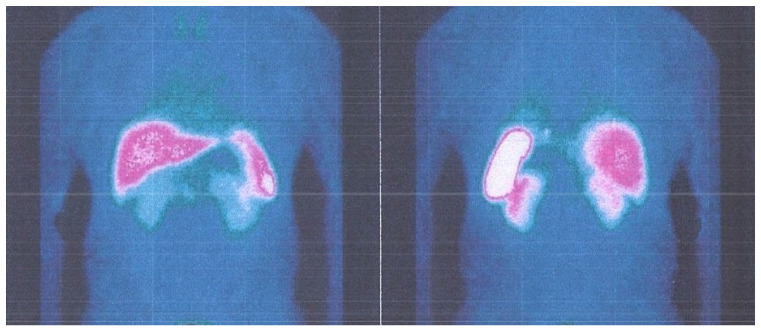
PET whole-body scanning showing the absence of tumoral relapse.

## Data Availability

Not applicable.

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
