# Peer review of "Duodenal Gangliocytic Paragangliomas—Case Series and Literature Review"

_life, 2023, doi:10.3390/life13030597_

Round 1

Reviewer 1 Report

In the present article, Stan-Ilie et al describe three cases of duodenal gangliocytic paragangliomas (GP), a rare form of neuroendocrine tumor.

This is a nice description of case series. Only a linguistic revision is necessary, see for example:

1) Page 2 line 51: vescicular? This sentence is strange.

2) Page 4 line 125: beforehand?

3) Page 9 line 277: it’s  - - > its.

4) Page 9 line 290: reccurence - - > recurrence

5) Page 10 line 321: viewing - - > viewing.

6) In figures 2-3, please add arrows to underline findings.

Reviewer 2 Report

Madalina Stan-Ilie et al. uncover Duodenal Gangliocytic Paragangliomas.

The topic is of interest.

Points to be improved:

1) The rationale of why the authors came up with this research is scanty and is related to a lack of novelty: please highlight what this manuscript might add.

2) What is the information that is not exactly available that motivated the authors to come up with this information. What are the current caveats and how do the authors highlight the current research in answering them? If not they need to address in background and infuture directions .

3)State of the art figures are required: scale bar should be provided in high resolution.

4)The authors could provide a little more consideration of genomic directed stratifications in clinical trial design and enrolments. 

5)The underlying message here is that more precision and individualized approaches need to be tested in well-designed clinical trials – a challenge, but I would be interested in their perspective of how this might be done. If beyond the scope of the manuscript, this should be highlighted as a limitation

6) The authors need to highlight what new information the review is providing to enhance the research in progress

7) I would suggest to slightly restructure the manuscript as follows:

Part 1 — Working Title, WHAT happened: Timeline and Narrative

Develop a descriptive and succinct working title that describes the phenomenon of greatest interest (symptom, diagnostic test, diagnosis, intervention, outcome).

WHAT happened. Gather the clinical information associated with patient visits in this this case report to create a timeline as a figure or table. The timeline is  a chronological summary of the visits that make up the episodes of care from this case report.

Narrative of the episode of care (including tables and figures as needed).

The presenting concerns (chief complaints) and relevant demographic information.

Clinical findings: describe the relevant past medical history, pertinent co-morbidities, and important physical examination (PE) findings.

Diagnostic assessments: discuss diagnostic testing and results, a differential diagnosis, and the diagnosis.

Therapeutic interventions: describe the types of intervention (pharmacologic, surgical, preventive, lifestyle) and how the interventions were administered (dosage, strength, duration, and frequency). Tables or figures may be useful.

Follow-up and outcomes: describe the clinical course of the episode of care during follow-up visits including (1) intervention modification, interruption, or discontinuation; (2) intervention adherence and how this was assessed; and (3) adverse effects or unanticipated events. Regular patient report outcome measurement surveys such as PROMIS® may be helpful.

Part 2 — WHY it might have happened: Introduction, Discussion, Conclusion

The introduction should briefly summarize why this case report is important and cite the most recent CARE article (Riley DS, Barber MS, Kienle GS, AronsonJK, et al. CARE guidelines for case reports: explanation and elaboration document. JClinEpi 2017 Sep;89:218-235. doi: 10.1016/jclinepi.2017.04.026).

WHY it might have happened. The discussion describes case management, including strengths and limitations with scientific references.

The conclusion, usually one paragraph, offers the most important findings from the case without references.

Part 3 — Abstract, Keywords, References, Acknowledgements, and Informed Consent

Abstract. Briefly summarize in a structured or unstructured format the relevant information without citations. Do this after writing the case report. Information should include: (1) Background, (2) Key points from the case; and (3) Main lessons to be learned from this case report.

Keywords. Provide 2 to 5 keywords that will identify important topics covered by this case report.

References. Include appropriately chosen references from the peer-reviewed scientific literature.

Acknowledgements. A short acknowledgements section should mention funding support or conflicts of interest, if applicable.

Informed Consent and Patient Perspective. The patient should provide informed consent (including a patient perspective) and the author should provide this information if requested. Some journals have consent forms which must be used regardless of informed consents you have obtained. Rarely, additional approval (e.g., IRB or ethics commission) may be needed. The patient should share their perspective on the treatment(s) they received in one to two paragraphs. It is often best to ask for informed consent and the patient’s perspective before you begin writing your case report.

Appendices (If indicated).

  • 8) This reviewer personally misses some insights regarding novel topics for paraganglioma:  antiangiogenic medications are currently being evaluated in prospective clinical trials for patients with metastatic pheochromocytomas and paragangliomas, and preliminary results have been encouraging. As is now well known, tumors grow and evolve through a constant crosstalk with the surrounding microenvironment, and emerging evidence indicates that angiogenesis and immunosuppression frequently occur simultaneously in response to this crosstalk. Accordingly, strategies combining anti-angiogenic therapy and immunotherapy seem to have the potential to tip the balance of the tumor microenvironment and improve treatment response (please refer to PMID: 34298648 and expand the introduction/discussion sections).

Round 2

Reviewer 2 Report

The authors have clarified several of the questions I raised in my previous review. Most of the major problems have been addressed by this revision. Nevertheless, the authors neglected some of the previous comments in my report. I still feel 27 references are scanty for a review, but the reviewing process is also subjective. The final decision is for the editor.

Author Response

Dear reviewer,

We want to thank you for all your suggestions and comments and for taking the time to review our paper.

We further investigated and added six more references. However, the literature is scarce regarding duodenal gangliocytic paragangliomas, therefore the list of references might be scanty.

Kind regards,

dr Oana Plotogea
